# Delirium in Palliative Care

**DOI:** 10.3390/cancers13235893

**Published:** 2021-11-23

**Authors:** Patricia Bramati, Eduardo Bruera

**Affiliations:** Department of Palliative Care, Rehabilitation and Integrative Medicine, The University of Texas MD Anderson Cancer Center, 1515 Holcombe Blvd., Unit 1414, Houston, TX 77030, USA; pbramati@mdanderson.org

**Keywords:** delirium, palliative, sedation, antipsychotics, haloperidol, benzodiazepines

## Abstract

**Simple Summary:**

Delirium is a generalized cerebral dysfunction that occurs frequently near the end of life. In palliative care, delirium is frequently a sign of impending death; it is distressing for patients, families, and caregivers; and the goals of management, assessment, and treatment are controversial. We provide an update on these topics mainly focusing on patients with cancer.

**Abstract:**

Delirium, a widespread neuropsychiatric disorder in patients with terminal diseases, is associated with increased morbidity and mortality, profoundly impacting patients, their families, and caregivers. Although frequently missed, the effective recognition of delirium demands attention and commitment. Reversibility is frequently not achievable. Non-pharmacological and pharmacological interventions are commonly used but largely unproven. Palliative sedation, although controversial, should be considered for refractory delirium. Psychological assistance should be available to patients and their families at all times.

## 1. Introduction

The word “delirium” has its roots in the Latin word “*delirare*”, which meant ‘to go out of the furrow’, and this could be interpreted as deviating from a straight line or to be deranged [1]. Delirium is a common neuropsychiatric complication in post-surgical patients, in patients with strokes, in the intensive care unit (in particular, in patients requiring mechanical ventilation) and at the of life, mainly in a patient with advanced cancer [1,2]. Frequently, delirium becomes the final struggle of palliative care management [3]. Given the diverse terminology relating to delirium and also the ambiguity in establishing the onset of the end of life, the epidemiology of this syndrome varies widely [2]. Nonetheless, the best evidence suggests that almost all the patients will have delirium in the hours to days preceding their demise [2]. In palliative care, delirium is a sign of impending death, and in this context, the anguish felt by patients, families, and caregivers is usually worsened by the patient’s impaired communication and by behavioral control issues [2]. Determining whether or not delirium is reversible, as well as whether or not a potential reversal strategy should be explored, remains a challenge [2]. Another difficulty is that given the limited evidence supporting the non-pharmacological and pharmacological treatments of delirium, the management remains controversial. Furthermore, ethical problems might arise when the only option becomes the administration of palliative sedation in order to control the symptoms of some patients [2]. In this review, we will provide an up-to-date summary of the diagnosis, assessment tools, epidemiology, etiology, and management of delirium in palliative care, with a particular focus on patients with cancer at the end of their lives.

## 2. Delirium Diagnosis and Clinical Features

Delirium is a global cerebral dysfunction with multiple symptoms and signs that constitute a neurocognitive or neuropsychiatric condition with impaired attention as the defining feature [4]. Not without criticism [5], the diagnostic criteria of delirium from the 5th edition of the Diagnostic and Statistical Manual of Mental Disorders (DSM-5) have become standard [6]. The essential feature for diagnosis is a disturbance in attention and awareness over a short period of time with fluctuating severity over the course of a day, as well as a disturbance in cognition (Table 1); importantly these changes cannot occur in the presence of a severely reduced level of arousal, such as coma [6].

Outside the United States, the criteria of the 11th edition of the International Classification of Diseases (ICD-11), by the World Health Organization are widely used [7]. The ICD-11 criteria overlap with the DSM-5 and add components of the sleep-awake cycle, as reduced arousal of acute onset, or total sleep loss with reversal of the sleep-wake cycle [7]. The ICD-11 criteria state that:


*“Delirium is characterized by a disturbance of attention, orientation, and awareness that develops within a short period of time, with transient symptoms that may fluctuate depending on the underlying causal condition or etiology. Delirium often includes disturbance of behavior and emotion and may include impairment in multiple cognitive domains. A disturbance of the sleep-wake cycle, including reduced arousal of acute onset or total sleep loss with reversal of the sleep-wake cycle, may also be present. Delirium may be caused by the direct physiological effects of a medical condition not classified under mental, behavioral, or neurodevelopmental disorders, or of a substance or medication, including withdrawal, or by multiple or unknown etiological factors.”*


The frequency of the symptoms of delirium varies: attention deficits are reported in 97 to 100% of the patients, thought process abnormalities in 54 to 79%, disorientation in 76 to 96%, memory deficits in 88 to 96%, sleep-wake disturbances in 92 to 97%, motor alterations in 24 to 94%, language disturbances in 57 to 67%, perceptual disturbances in 50 to 63%, delusions in 21 to 31% and affective changes in 43 to 86% [8].

Delirium has been categorized into three clinical subtypes: hypoactive, hyperactive, and mixed [8]. In hypoactive delirium, the patients look lethargic and drowsy, respond to questions slowly, do not initiate movements, and have a diminished awareness of their surroundings [9]. The most common features of the hyperactive subtype are restlessness, agitation, hypervigilance, hallucinations, and delusions [9]. Patients with the mixed subtype exhibit alternating aspects of hypoactive and hyperactive subtypes [10]. Hypoactive delirium is not infrequently confused with fatigue, depression, or dementia, and hyperactive delirium with anxiety or psychosis [11]. Fluctuations in the signs or symptoms can result in lucid intervals which might coincide with the physician’s assessment resulting in a missed diagnosis [2].

## 3. Delirium Assessment Tools

The criteria for delirium based on the DSM-5 or ICD-11 (or their previous versions) are used in clinical practice and research. Usually, the routine examination of patients is not adequate to identify delirium so different tools have been developed to formally standardize the assessment. Over fifty delirium assessment tools have been created [12,13,14,15,16] for different purposes, such as the intermittent utilization at the initial or later encounters, to monitor for new-onset delirium, for ultra-brief screening, for a detailed phenomenological and/or neuropsychological assessment, and for the measurement of the severity of delirium [1].

The Confusion Assessment Method (CAM) [17], which is a copyrighted tool available from the website of the Hospital Elder Life Program (HELP) [18], is the most widely utilized instrument in published investigations [1], it is also frequently employed in clinical practice [1] and has been validated in palliative care [19]. The full version of the CAM has nine components based on DSM-III R criteria. The CAM diagnostic algorithm consists of four items: acute onset and fluctuating course (criterion A), inattention (criterion B), disorganized thinking (criterion C), and level of consciousness (criterion D) [1,4]. Trained evaluators administer a cognitive test when meeting the patient, and then classify each item as present or absent, with the complete examination taking 5 to 10 min. For the CAM to be positive, criteria A and B need to be present as well as either or both criteria C and D [1,4]. Several variants of the CAM exist as the brief CAM (bCAM), 3D-CAM, or the CAM-ICU [1].

There are different tools to assess the severity of delirium which look at the magnitude of the cognitive impairment, the level of arousal, the duration, the number of present criteria, and the level of distress. Two of the most frequently used instruments, which have been validated in patients with advanced cancer, are the Memorial Delirium Assessment Scale (MDAS) [20] and the Delirium Rating Scale—Revised 98 (DRS-R98) [21].

The MDAS has ten components which are graded from 0 to 3, with a total range of 0 to 30, and it is used to measure the severity of delirium. The MDAS was created using the DSM IV delirium criteria as well as delirium symptoms from previous or alternative classification systems (such as the DSM III, DSM III-R, and ICD-9) [20]. Although originally designed to evaluate severity, the MDAS has also been validated as a diagnostic tool when using a threshold score of ≥7 [22]. In our palliative care inpatient unit, the MDAS is obtained in every patient on a daily basis which allows the continuous assessment of the severity of delirium.

The DRS-R98 has 16 components of which 13 are for severity and 3 for diagnosis with a maximum score of 46 (including the diagnostic components) and maximum severity of 39. It normally takes between 20 and 30 min to complete. Three words are used to evaluate short-term memory; to grade attention, the months of the year backward are utilized; to evaluate visual constructional skills the patient is asked to draw a clock face, and parts of a pen and or watch are used to evaluate naming. When using a cutoff score of 15.25, the sensitivity and specificity of the DRS-R-98 severity scale was 92% and 93% respectively. The DRS-R-98 is an excellent tool for long-term research [21].

## 4. Delirium Epidemiology and Impact

Delirium is the most prevalent neurological problem in patients with advanced illness [23], but the reported epidemiology in patients with advanced cancer varies greatly depending on the study population, the definition, and the methodology of evaluation [24]. The hypoactive subtype is the most frequent subset found in patients with cancer and it is commonly missed by the treating team [25]. To demonstrate the latter point, de la Cruz et al. showed that the primary medical team in a tertiary oncological medical center missed 61% of patients with delirium who were diagnosed by a palliative care specialist [26]. The reported prevalence of delirium also varies depending on the patient setting; in an emergency room, delirium was encountered in about 10% of advanced cancer patients, while it was present in 43% and 42% of admissions to a general medical ward and a palliative care unit, respectively. At the end of life, about 90% of the patients who died in an acute palliative care unit had delirium [25].

Delirium predicts death in terminally ill patients within days to weeks, in particular in patients with advanced cancer in palliative care units and the hospice settings. Importantly, if advance care planning has not occurred prior to an episode of delirium in a terminally ill patient, it is frequently too late to do so [3].

Delirium causes psychological stress in the patients who survive, in family members, and the treating team. Breibart et al. [27] and Bruera et al. [28] found that 54% and 74% of surviving patients with delirium respectively recalled the experience after recovering [27]. Patients with delirium recall experienced severe distress as compared to those with no recall [28]. The more severe the delirium episode, the less likely the patient will recall it; however, if hallucinations or delusions are present, the likelihood of remembering the episode increases.

## 5. Etiologies and Diagnosis of Precipitants of Delirium

Episodes of delirium have been reported to be reversible between 25% and 68% [29,30], so it is important to try to identify the precipitants and correct them if possible. However, an etiology is discovered in less than half of terminally ill patients [31]. So, given the poor prognosis of delirium in palliative care, an individualized and careful search should be aligned with the goals of care. Figure 1 shows the main factors contributing to delirium in cancer patients [32]. A careful history should look for adverse effects from medications such as opioids, corticosteroids, benzodiazepines, neuroleptics, etc. [25]. Of particular importance is the potential for opioid toxicity, and the approach may include a rotation of the opioid or a reduction of the dose. Family members usually prefer changing the opioid to a new one, rather than reducing the dose, when considering the diagnosis of delirium caused by opioid toxicity, especially in agitated patients [2]. Polypharmacy should be addressed and avoided. Laboratory tests can identify electrolytes abnormalities as hypercalcemia, hyper/hyponatremia, hypomagnesemia, and can also help with the diagnosis of hypoglycemia, hypoxemia, uremia, hyperammonemia, etc. Imaging can identify pneumonia, brain metastasis, or leptomeningeal carcinomatosis. Additionally, dehydration and alcohol withdrawal should be considered and treated if present [3,25,32].

The pathophysiology of delirium is beyond the scope of this review and has been exhaustively reviewed elsewhere [33]. Nonetheless, Maldonado has proposed “the *systems integration failure hypothesis*” in which changes in the synthesis of neurotransmitters and their availability, combined with a failure of interconnected brain systems, results in a failure of the functional integration and processing of information of the central nervous system [33]. Multiple factors have been implicated in this hypothesis, including neuroinflammation, brain vascular dysfunction, altered brain metabolism, neurotransmitter imbalance, and impaired neuronal network connectivity [1]. As a consequence, low levels of acetylcholine and high levels of dopamine with alterations in glutamate and gamma-aminobutyric acid are encountered in patients with delirium [34]. Interestingly, it has been shown in rats that the injection of morphine in the hypothalamic paraventricular nucleus increases the levels of dopamine and decreases the levels of acetylcholine [35], providing a potential explanation for opioid-induced delirium -which represents about half of the cases of opioid-induced neurotoxicity in palliative care (about 15% of palliative care patients experience opioid-induced neurotoxicity) [36]. Additionally, patients with advanced cancer might have a dysregulation of the inflammatory pathways and cytokines (also known as the *neuroinflammatory hypothesis*) leading to abnormal activation of the central nervous system altering the function of synapses [34]. Finally, patients with advanced cancer might experience an increase in oxidative stress (also known as the *oxidative stress hypothesis*) causing cerebral damage [34].

## 6. Management of Delirium

The main goal of delirium management in the palliative care setting, given its poor prognosis, is the comfort of the patient. Ideally, the patient should be awake, alert, calm, not in pain, cognitively alert, and capable of communicating with his family and staff [3]. The management plan should consider the following problems: the safety of the patient and those around the patient, the best location of care (home, hospital, palliative care unit, etc.), information and education for the patients and their family, correction of underlying causes if possible, minimization of offenders which may aggravate symptoms (urinary retention, constipation, pain, etc.), avoidance of dehydration, determination of mediators of distress, consideration of patient priorities, understanding that for the family delirium is as a loss of the person they knew and awareness of the poor prognosis of the patient [11].

Agitation is frequently misjudged as pain by relatives and the staff. The psychological trauma experienced by families during an episode of agitation caused by delirium can cause a profound misperception of the problem which has been referred to by Fainsinger et al. as the “destructive triangle” [37], in which the patient, the families, and the caregivers become increasingly distressed. As a consequence, the relatives pressure the nurses and healthcare staff to ramp up sedation and/or analgesics to alleviate the perceived misery of their loved one, which might result in the inappropriate escalation of opioids leading to toxicity [2].

Unless family members comprehend that the agitation is a symptom of delirium, their perception will be that the patient is experiencing unbearable pain. If the agitation is not managed successfully, the family may be left with the horrifying memory of a loved one dying in agony. This memory may later complicate the bereavement of the surviving relatives [2]. Ironically, one of the most significant advantages of controlling refractory delirium is the documented relief and gratitude of the families [30].

### 6.1. Non-Pharmacological Interventions

Several non-pharmacological interventions for delirium management and prevention have been recommended in various clinical practice guidelines, targeting patients in different locations, including hospitals, long-term care institutions, hospices, and other palliative care settings [25]. A list of these interventions is presented in Table 2. A metanalysis of non-pharmacological interventions by Hshieh et al. from 2015 (not in palliative care) which included 14 studies found a 44% decrease in the incidence of delirium, a 64% decrease in the rate of falls, and a trend towards a small decrease in the length of stay and the necessity for institutionalization [38]. However, in a more recent systematic review, Hosie et al. demonstrated that non-pharmacological strategies have frequently excluded patients in need of palliative care and the outcomes were rarely reported [39]. Furthermore, the European Society of Medical Oncology (ESMO) stated in their practice guidelines for delirium in cancer patients, that most non-pharmacological interventions have no evidence to substantiate its use [25]. In coincidence with these latter two opinions, a recent stepped-wedge cluster randomized control study which was conducted in ten intensive care units (this study was not in the palliative care setting) using a multicomponent nursing intervention centered on enhancing vision, hearing, orientation, cognition, and mobility found no change in the number of delirium-free and coma-free days within 28 days following the admission to the intensive care unit or in any of the secondary outcomes [40].

Nevertheless, adopting some of these interventions such as avoiding or treating dehydration, encouraging mobility and ambulation, and trying to improve the orientation of the patients, seems appropriate given their low risk and their potential benefit regardless of delirium [41]. It is important to note that one-to-one nursing care might be required, and physical restraints should be avoided. Restraints should be used only when a patient poses a clear risk of harm to himself/herself or others and no other option is available; restraint orders should always be time-limited, and their necessity should be reassessed frequently [3]. Additionally, decreasing the utilization of restraints has been linked to better outcomes in delirium-prone patients [42]. A different type of restraint is bed and chair alarms which are used to prevent falling and to limit the mobility of unaccompanied patients, however, these alarms have not been shown to be beneficial in reducing the risk of falling and might create further stress to an already agitated patient [43].

### 6.2. Pharmacological Management of Delirium

The imbalance of neurotransmitters, resulting in a surplus of dopaminergic and deficit of cholinergic transmission, has provided the historic rationale for the use of antipsychotics (dopamine D_2_ receptor antagonists) in delirium over the last two decades [25,33]. The most common antipsychotic drugs used include phenothiazines such as chlorpromazine, butyrophenones such as haloperidol, second-generation antipsychotics such as olanzapine, quetiapine, risperidone, and ziprasidone, and third-generation antipsychotics such as aripiprazole [44]. Despite little supporting data, and that there is no approved medication to treat delirium, antipsychotics are widely used [45].

Outside the palliative care specialty, the role of antipsychotics remains uncertain [45]. Studies of antipsychotics for managing delirium suggest that they improve delirium severity [46,47,48,49,50,51,52]; however, all the studies have significant shortcomings: in one the concealment of allocation was imperfect [49], most of them were underpowered [46,47,48,49,50,51,52], and only three were placebo-controlled trials [49,50,52]. Moreover, two studies of antipsychotics in critically ill adults were negative [53,54].

The most recent Cochrane review of drug therapy for delirium in terminally ill adults released in 2020 [55], included only four studies [45,46,56,57]. The included studies needed to meet the following strict criteria: randomized controlled trials of drugs (in any dose by any route) compared to another medication, to a non-pharmacological strategy, to placebo, to standard care or a wait-list control, for the management of delirium in terminally ill adults, from its inception to July 2019.

Going in chronological order of publication, the first study included in the Cochrane review was conducted by Breibart et al. and evaluated haloperidol (*n* = 11), chlorpromazine (*n* = 13), and lorazepam (*n* = 6) for the treatment of delirium in patients with human immunodeficiency virus (HIV) [46]. Although not in cancer, the 30 patients included had advanced disease requiring medical treatment. Using the Delirium Rating Scale to measure the intensity of delirium, haloperidol, and chlorpromazine (in relatively low doses) resulted in a significant improvement of the symptoms. Remarkably, all the patients treated with lorazepam suffered treatment-limiting adverse effects (over sedation and increased confusion), prompting the authors to discontinue the lorazepam arm. Because of its small size and the unique medication schedule, this study should be read with caution [23]. In the second study included in the review, which was conducted by Lin et al. [57], 30 patients with advanced cancer undergoing hospice or palliative care, were given olanzapine (*n* = 16) or haloperidol (*n* = 14). Both groups showed delirium improvement with no significant difference between them. However, the small size of this study, as well as the lack of a placebo arm, obscures the interpretation of the results.

The third study included in the Cochrane review is the one by Agar et al. [45]. This study (the largest yet) randomized (intention to treat) 247 patients (218 had cancer) to risperidone (*n* = 82), haloperidol (*n* = 81), or placebo (*n* = 84) in a double-blind, parallel-arm, dose-titrated randomized trial conducted at eleven Australian inpatient hospice or hospital palliative care services. Remarkedly, the primary outcome, a composite score based on inappropriate behavior, inappropriate communication, and illusions/hallucinations was significantly worse in the risperidone and haloperidol arms compared to the placebo arm [45]. Patients in both treatment arms experienced more extrapyramidal side effects compared to placebo, and unexpectedly participants in the placebo group had better overall survival than those treated with haloperidol (hazard ratio, 1.73; 95% CI, 1.20–2.50; *p*  = 0.003), although this was not significant when placebo was compared with risperidone (hazard ratio, 1.29; 95% CI, 0.91–1.84; *p*  = 0.14) [45]. The interpretation of some of these findings is complicated by the use of a non-validated primary outcome, enrollment of patients with mild delirium, non-exclusion of patients with dementia, and the use of low-dose oral neuroleptics [23].

One important aspect of the fourth study included in the Cochrane review, by Hui et al. [56], is that rather than looking for the improvement of delirium, the objective was the control of agitation (probably the most troubling feature of delirium). This single-center, double-blind, randomized controlled trial compared haloperidol plus lorazepam (*n* = 47) versus haloperidol plus placebo (*n* = 43) in patients hospitalized with advanced cancer and agitated delirium. The Richmond Agitation Sedation Scale (RASS) was used to assess the agitation [58]. The haloperidol plus lorazepam arm significantly reduced the RASS score at 8 h more than the haloperidol plus placebo arm [56]. Furthermore, patients in the haloperidol plus lorazepam arm required fewer rescue neuroleptics and were perceived to be more comfortable by caregivers and nurses [56]. Since haloperidol was included in both arms, the group differences can be attributed to the benzodiazepine, and importantly the rate of adverse effect and overall survival was not significantly different between arms. Overall survival was only three days, highlighting the poor prognosis of this population of patients [56].

The Cochrane review concluded that there is no high-quality evidence supporting or refuting the use of pharmacological therapies for delirium in terminally ill adults; low-quality evidence was found that risperidone or haloperidol might slightly aggravate delirium compared to placebo and might slightly increase extrapyramidal adverse events in people with mild to moderate delirium [55]. Finally, because of the small number of studies (four) and participants (399 adults in total, of whom about 90% were oncological patients), further research is essential [55].

A more recent study [59], which was not included in the Cochrane review, compared three different strategies for managing refractory delirium in patients admitted to a palliative and supportive care unit in a single tertiary oncological medical center. The authors compared the dose escalation of intravenous haloperidol, the neuroleptic rotation of haloperidol and chlorpromazine, or the combination of haloperidol and chlorpromazine until death or discharge. The three strategies in this small study (escalation *n* = 15, rotation *n* = 16, combination *n* = 14), which did not include a placebo group, were equally effective in reducing agitation [59].

When managing a patient with delirium, if pharmacological treatment is needed, a short-term, low-dose antipsychotic is recommended for symptoms of perceptual disturbance, to control severe agitation, or if there are safety concerns [11]. Patients should be constantly supervised to see if the medication is effective in alleviating either delirium or agitation [11]. A benzodiazepine may be indicated for severe agitation or alcohol withdrawal [23].

Palliative sedation may be required as the only way to control symptoms in patients with refractory agitated delirium. Although beyond the scope of this manuscript, refractory delirium and refractory dyspnea are the most common conditions which might require palliative sedation. Several medications (benzodiazepines, propofol, dexmedetomidine, etc.) can be used to achieve sedation in this difficult group of patients and palliative sedation remains a controversial intervention [60,61,62]. As an example, a study looked at all the admission to a palliative care unit of a tertiary oncological medical center for 2004 and 2005, 1207 admissions were identified of which 186 (15%) received palliative sedation [61]. The most common indications for palliative sedation were refractory delirium in 153 patients (82% of the cases) and refractory dyspnea in 11 patients (6% of the cases) [61].

When considering the use of palliative sedation, it is critical to be informed about the norms and laws of the country concerning end-of-life issues because there might be significant differences across nations and even between states within a single country. Several practice guidelines provide recommendations about this topic [63].

## 7. Conclusions

Delirium is a common neuropsychiatric condition that is associated with increased morbidity and mortality in patients with advanced illness [26], and profoundly impacts the patients, their families, caregivers, and staff. Although often missed, effective recognition demands attention and commitment. Caregivers taking care of the patient with advanced cancer should be aware of this condition and should be familiar with the available diagnostic tools used to identify delirium. Reversibility is frequently not achievable so symptomatic management with non-pharmacological interventions supplemented, if needed, with pharmacological interventions should be attempted. When everything else fails, palliative sedation should be considered. Psychological support should always be accessible to the patient and their families. More research is needed to establish a solid foundation for the management of delirium.

## Figures and Tables

**Figure 1 cancers-13-05893-f001:**
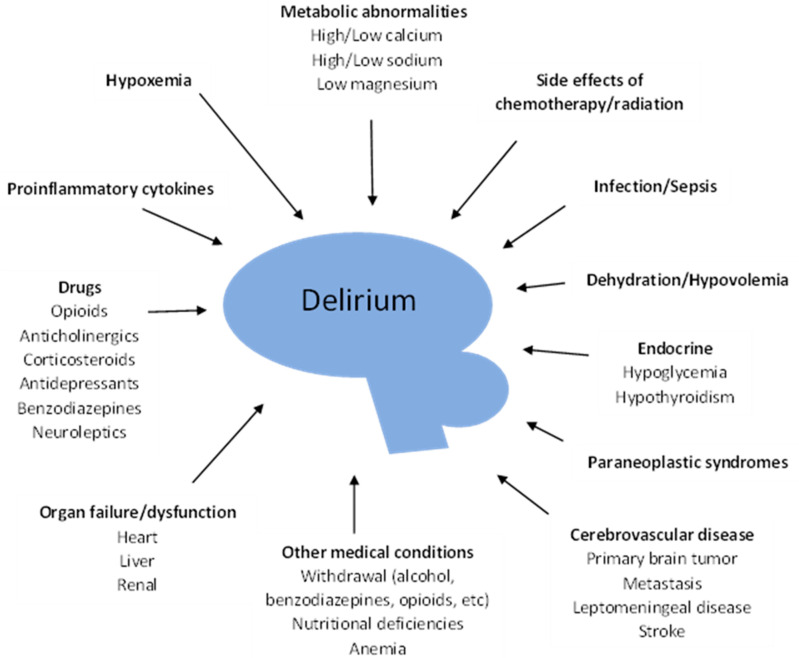
Factors associated with delirium in cancer patients [32].

**Table 1 cancers-13-05893-t001:** Delirium criteria are based on the fifth edition of the Diagnostic and Statistical Manual of Mental Disorders (DSM-5) [6].

A. Disturbance in attention (i.e., reduced ability to direct, focus, sustain and shift attention) and awareness (reduced orientation to the environment).
B. The disturbance develops over a short period of time (usually hours to a few days), represents a change from baseline attention and awareness, and tends to fluctuate in severity during the course of a day.
C. An additional disturbance in cognition (e.g., memory deficit, disorientation, language, visuospatial ability, or perception).
D. The disturbances in Criteria A and C are not better explained by a pre-existing, established, or evolving neurocognitive disorder and do not occur in the context of a severely reduced level of arousal, such as a coma.
E. There is evidence from the history, physical examination, or laboratory findings that the disturbance is a direct physiological consequence of another medical condition, substance intoxication or withdrawal (i.e., due to a drug of abuse or to a medication), or exposure to a toxin, or is due to multiple etiologies.

**Table 2 cancers-13-05893-t002:** Non-pharmacological interventions for delirium management. Adapted from Breitbart and Alici [3].

Minimize the use of immobilizing catheters, intravenous lines, and physical restraints
Avoid immobility, early mobilization
Monitor nutrition
Provide visual and hearing aids
Monitor closely for dehydration
Control pain
Monitor fluid-electrolyte balance
Monitor bowel and bladder functioning
Review medications
Reorient communications with the patient
Place orientation board, clock, or familiar objects (family photos, etc) in the room
Encourage cognitively stimulating activities such as word puzzles
Facilitate sleep hygiene, including relaxation music at bedtime, warm drinks, and gentle massage
Minimize noise and interventions at bedtime (reschedule medications, do not interrupt sleep for checking temperature, etc)

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
