# Peer review of "Delirium in Palliative Care"

_cancers, 2021, doi:10.3390/cancers13235893_

Round 1

Reviewer 1 Report

Thank you for the opportunity to review this manuscript.  The topic is of significant importance to  those working in palliative care who manage this distressing symptom on a regular basis.  It will also be of interest to other clinicians who may also be called on to manage delirium.

Author Response

We would like to thank the reviewer for his/her comments.

Reviewer 2 Report

This review is very well-written, comprehensive, and informative. I would recommend outlining at the beginning what the review will cover - later on you mention that certain topics are beyond the scope of the review but I was left wondering what the exact scope was. In addition, please consider outlining the unique contribution to the literature your manuscript is making.

I have picked up on some very minor typos as follows:

P2. Line 72 – insert “to” before questions.

p.4 line 147, add clarity to the following sentence: you write: Delirium has been reported to be reversible between 25% and 68% do you mean between 25% to 68% of cases?

p.4 line 158 insert “s” after test (Laboratory tests)

Line 179 insert “s” after patient

p.7 line 253 consider replacing “specially” with “specialty”

Author Response

We would like to thank the reviewer for his/her comments. We have changed the last sentence of the introduction outlining the topics to be covered in the review. We have also corrected all the typos and clarify the sentence in page 4, line 147 as recommended by the reviewer.

Reviewer 3 Report

The review is well conducted and addresses an important topic in the field.

The authors may want to elaborate on the specific practical implications in more detailed examples.

What are the biggest challenges regarding delirium during care in terminal life phases, for the patient, but also for the care staff and family/friends?

Which types of care seem most effective? To which extent such care needs to be tailored to the patient?

Author Response

We would like to thank the reviewer for his/her valuable comments.

Considering the first and third point raised by the reviewer given the limited evidence supporting any intervention to treat delirium, in particular in the palliative care setting, we feel that we will need so many examples to demonstrate different points that the review will end being a collection of case reports. And again, given the lack of evidence is difficult to suggest which type of care is most effective.

Regarding the second question raised by the reviewer, we think that we have addressed it in the last two paragraph of “Management of delirium” immediately before “Non-pharmacological interventions”. Nonetheless we have added the following sentence (in red) “Ironically, one of the greatest benefits of controlling refractory delirium is the documented relief and gratitude of the families [30].”